# Nanomedicine in the Treatment of Diabetes

**DOI:** 10.3390/ijms25137028

**Published:** 2024-06-27

**Authors:** Aikaterini Andreadi, Pietro Lodeserto, Federica Todaro, Marco Meloni, Maria Romano, Alessandro Minasi, Alfonso Bellia, Davide Lauro

**Affiliations:** 1Section of Endocrinology and Metabolic Diseases, Department of Systems Medicine, University of Rome Tor Vergata, 00133 Rome, Italy; lodesertopietro17@gmail.com (P.L.); fedetod@hotmail.it (F.T.); meloni.marco@libero.it (M.M.); bellia@med.uniroma2.it (A.B.); d.lauro@med.uniroma2.it (D.L.); 2Division of Endocrinology and Diabetology, Department of Medical Sciences, Fondazione Policlinico Tor Vergata, 00133 Rome, Italy; romanomaria76@gmail.com (M.R.); alessandrominasi87@gmail.com (A.M.)

**Keywords:** type 1 diabetes mellitus, insulin pathway, nanomedicine, oral formulation, nanoparticles, wound healing

## Abstract

Nanomedicine could improve the treatment of diabetes by exploiting various therapeutic mechanisms through the use of suitable nanoformulations. For example, glucose-sensitive nanoparticles can release insulin in response to high glucose levels, mimicking the physiological release of insulin. Oral nanoformulations for insulin uptake via the gut represent a long-sought alternative to subcutaneous injections, which cause pain, discomfort, and possible local infection. Nanoparticles containing oligonucleotides can be used in gene therapy and cell therapy to stimulate insulin production in β-cells or β-like cells and modulate the responses of T1DM-associated immune cells. In contrast, viral vectors do not induce immunogenicity. Finally, in diabetic wound healing, local delivery of nanoformulations containing regenerative molecules can stimulate tissue repair and thus provide a valuable tool to treat this diabetic complication. Here, we describe these different approaches to diabetes treatment with nanoformulations and their potential for clinical application.

## 1. Introduction

Diabetes mellitus is a metabolic disorder characterized by elevated blood glucose levels driven by insulin deficiency or resistance [1]. Type 1 diabetes (T1DM), also known as juvenile diabetes, accounts for ~10% of the diabetic patient population; it is associated with an insulin deficiency [2] caused by the autoimmune destruction of islet β-cells in the pancreas [3]. Type 2 diabetes (T2DM) occurs due to insulin resistance, a pathological condition characterized by the inability of the cells to respond to insulin or the downregulation of insulin receptors in response to hyperinsulinemia [4]. Persistent glycemic control is essential for both type 1 and type 2 diabetes. Blood glucose levels should be maintained within healthy normoglycemic ranges (70–140 mg per dl or 4–8 mM, known as euglycemia). 

When left untreated, prolonged hyperglycemia can lead to blindness, kidney and heart disease, nerve degeneration, and increased susceptibility to infection [5].

Anti-diabetic agents such as sulfonylureas (glipizide, glyburide, gliclazide, glimepiride), meglitinides (repaglinide and nateglinide), biguanides (metformin), thiazolidinediones (rosiglitazone, pioglitazone), α-glucosidase inhibitors (acarbose, miglitol, voglibose), DPP-4 inhibitors (sitagliptin, saxagliptin, vildagliptin, linagliptin, alogliptin), SGLT2 inhibitors (dapagliflozin and canagliflozin), and cycloset (bromocriptine) are currently used in the management of T2DM. They are administered via the oral route and generally provide good response rates.

In T1DM, on the contrary, lifelong insulin therapy is required. This could expose the patient to insulin overtreatment, resulting in hypoglycemia and the possibility of developing seizures, unconsciousness, or death [6]. Conventional therapies for both types of diabetes typically consist of frequent glucose monitoring and insulin administration (e.g., through subcutaneous injections or insulin pumps) throughout the day. However, sporadic glucose monitoring and poor patient adherence, driven by a multitude of factors, including pain and the tediousness of procedures [1], often lead to unpredictable insulin dosages, which could result in uncontrolled hyperglycemia, hypoglycemia, seizures, unconsciousness, or death [6]. Although continuous glucose monitors and insulin pumps have been developed to address these issues, and recent “closed-loop systems” consisting of pumps that deliver appropriate insulin amounts in response to blood glucose levels have been introduced, glucose management is still unsuccessful [7,8]. Therefore, there is a need to improve diabetes management. Recent advances in nanomedicine have significantly improved the treatment of medical conditions [9,10,11] such as cancer, autoimmune disorders, inflammatory and fibrotic diseases [12], and heart diseases [13]. They are now actively studied in diabetes management [14,15]. Nanomedicine mainly involves the application of nanoparticles loaded with drugs capable of controlling their release through different mechanisms influenced by environmental conditions. Controlled drug release has been shown to provide targeted therapies, precise drug delivery, and improved therapeutic outcomes [14]. In the management of diabetes, nanoparticles have different purposes, mainly to provide insulin release in response to changes in glucose concentration, thus reducing the risk of hypoglycemia [16].

Another purpose of theirs is to protect insulin from degradation in the gastrointestinal environment, thus allowing oral administration and avoiding daily injections. Furthermore, nanomedicine can be used in gene and cell therapy for diabetes. Nanoparticles containing oligonucleotides for the expression of insulin genes or the repression of defective genes can be used as non-viral vectors for transducing pancreatic cells or stem cells both in vitro and in vivo [17,18]. Nanomedicine can also be exploited in regenerative medicine, which in diabetes represents a precious tool for managing complications such as diabetic wounds. This review will focus on the recent applications of nanomedicine in diabetic therapies, their developments, and the possibility of clinical translation.

## 2. Main Types of Nanoparticles Used in the Management of Diabetes

Nanoparticles are biocompatible and biodegradable spherical systems containing conventional or biological drugs such as peptides and oligonucleotides. Nanoparticles are small on the nanometric scale, generally in the range of 100–300 nm. Oral or parenteral routes can be used for their administration. They behave as drug carriers that can protect the drug from the environmental conditions at the administration site, transport their drug cargo to target body compartments, and release the drug under environmental stimuli at the target site.

They can also provide non-viral cell transfection, both in vivo and in vitro, by transporting oligonucleotides to target cells to correct defective gene expression. The main types of nanoparticles used in diabetes treatment are polymeric nanoparticles, polymeric nanocapsules, liposomes, and lipid nanoparticles [19] (Figure 1).

### 2.1. Polymeric Nanoparticles

Polymeric nanoparticles consist of solid polymer matrices containing uniformly dispersed drugs. Polymers can be natural, such as polysaccharides (chitosan, sodium alginate, hyaluronic acid) and proteins (serum albumin, gelatin, keratin), or synthetic, such as polylactic acid (PLA), poly lactic-co-glycolic acid (PLGA), polyethyleneimine, etc.

Drug release occurs through the destabilization of these matrices in the body’s fluids. Hydrophilic matrices based on polysaccharides and proteins are destabilized by the rapid penetration of water and subsequent swelling. They are suitable for releasing hydrophilic molecules such as insulin and other biological drugs. Hydrophobic matrices based on PLA or PLGA are destabilized by biodegradation following the slow penetration of water. They are used for both hydrophilic and lipophilic drugs when prolonged drug delivery over time is desired [20].

### 2.2. Polymeric Nanocapsules

Polymeric nanocapsules are characterized by solid polymer shells that enclose internal liquid phases. The polymeric shell may contain proteins such as serum albumin, which must be cross-linked to avoid solubilization in the aqueous bodily fluids in the administration environment. Albumin nanocapsules contain hydrophobic liquids such as vegetable oils that are suitable for encapsulating lipophilic drugs.

When the polymer shell is made of PLA or PLGA, the internal liquid phase of the nanocapsules is aqueous. Therefore, they are suitable for encapsulating hydrophilic drugs such as insulin and other biologics [21]. 

### 2.3. Liposomes

Liposomes are vesicles composed of phospholipid bilayers that enclose an aqueous phase. Liposomes are suitable for hydrophilic drugs solubilized in water, and hydrophobic medicines solubilized in the lipid bilayer [22]. The appropriate choice of phospholipid types and the mixing of phospholipids with other lipid molecules can modify the physicochemical characteristics of the liposome surface, thus modifying the release characteristics across different administration sites and body compartments. 

### 2.4. Lipid Nanoparticles

Lipid nanoparticles are liposome-like structures widely used for the delivery of proteins and oligonucleotides. They differ from liposomes because they consist of a single phospholipid layer, containing inverted micellar structures within the nanoparticle core. These inverted micelles have the ability to entrap hydrophilic drugs inside the nanoparticles, thus providing protective carrier systems for structures such as proteins or oligonucleotides that can be easily degraded in bodily fluids. The morphology of lipid nanoparticles and their drug loading can be modified by varying the physicochemical and formulative parameters [23].

## 3. Nanoparticles for Releasing Insulin in Response to Blood Glucose Levels

Nanoparticles responsive to blood glucose levels are designed to mimic the physiological response to changes in blood glucose concentration and modulate insulin release accordingly. The aim is to provide tighter glycemic control than conventional subcutaneous injections, minimizing the potential for hypoglycemia. Glucose-mediated insulin release relies on several mechanisms, the main of which is based on the ability of phenylboronic acid (PBA) to form reversible bonds with glucose molecules. 

Nanoparticles containing polysaccharides and PBA-conjugated polymers are characterized by multiple interactions between glucose and PBA that stabilize their matrix. In the presence of high glucose levels in bodily fluids, the competitive binding of the environmental glucose, which penetrates the matrix, to PBA determines the matrix’s destabilization and the consequent release of insulin (Figure 2a). Nanoparticles prepared from alginate, 3-aminoPBA, chitosan 3-fluoro-4-carboxylPBA, and mesoporous silica have good properties as glucose-responsive oral insulin-delivery systems [24].

Other systems have been prepared using PBA linked to proteins and glucose-decorated polymers [25,26]. One problem with PBA is that it does not have a great specificity for glucose, as it has the ability to interact with diols of molecules other than glucose present in the body, such as fructose, catechols, nucleic acids, and low-molecular-weight proteins. Efforts are underway to develop PBA derivatives that may have a greater specificity for glucose at physiological pH [27,28].

Another glucose-responsive release mechanism uses glucose-binding proteins such as concanavalin A (ConA) to form supramolecular complexes with glucose. Multiple inter-polymer interactions stabilize nanoparticles prepared from polysaccharides and polymers linked to ConA. In the presence of high glucose levels in bodily fluids, the competitive binding of environmental glucose, which penetrates the matrix, to ConA induces matrix destabilization and insulin release [29,30] (Figure 2b).

Other glucose-responsive nanoparticles rely on the glucose-sensitive enzyme glucose oxidase, which enzymatically converts glucose to gluconic acid, generating an acidic microenvironment. Nanoparticles containing glucose oxidase and pH-sensitive polymers can release insulin in response to hyperglycemic conditions. The acidic protonation of these pH-sensitive polymers triggers matrix hydration, swelling, and disassembly, with a consequent insulin release (Figure 2c).

The conversion of glucose to gluconic acid [31] by glucose oxidase is the most studied of the various glucose-sensing mechanisms described in the literature [32], and it is currently used in glucose sensors.

Many preclinical studies have been carried out on these systems. pH-sensitive polymers such as chitosan [33], poly[2-(dimethylamine) ethyl methacrylate] [34], poly(ethylenimine), and acetal-modified dextran have been used [35]. Good results have been obtained in response to changes in glucose concentrations, and euglycemia in rodents has been achieved with these systems [32]. However, the enzymatic conversion of glucose is still unreliable due to changes in glucose concentrations at different points of the matrix, which affect the decrease in pH and the consequent insulin release [36]. Currently, only preclinical studies have been carried on insulin therapeutics whose bioactivity is regulated by blood glucose levels. Further research is needed to make these systems suitable for entering clinical trials.

## 4. Nanoparticles for the Oral Administration of Insulin

Oral administration is undoubtedly the ideal route for administering drugs, with the advantages of simplicity, practicality, and good patient compliance. In the case of insulin, in particular, oral administration is considered the most advantageous in treating diabetes because it closely mimics endogenous insulin secretion. Endogenous human insulin secreted by the pancreas reaches the liver via the portal vein and then enters the systemic circulation. Of the insulin entering the liver, up to 80% is trapped by first-pass metabolism, resulting in a significantly higher insulin concentration in the portal vein than in the systemic circulation [37]. The remaining insulin enters the circulation and is distributed to various body compartments. This portal–peripheral gradient, characterized by a much higher concentration of insulin in the liver than in other organs and tissues, controls the extraction of insulin from the liver and maintains the optimal levels of peripheral insulin [37] (Figure 3).

Insulin administered via the oral route is absorbed through the intestinal mucosa and transported to the liver via the portal vein, thus establishing a portal–peripheral insulin gradient similar to physiological insulin secretion (Figure 3). Conversely, insulin administered via subcutaneous injection first enters the peripheral systemic circulation and then enters the liver, which can easily cause peripheral hyperinsulinemia.

Therefore, the development of oral insulin formulations could provide practical therapeutic tools in the treatment of diabetes. However, the oral absorption of insulin, like other biological drugs, is well known to be prevented by various physiological barriers, including denaturation in the gastric acidic environment, degradation by gastrointestinal proteases, and hindered absorption through the intestinal epithelium. Using enteric-coated formulations, enzyme inhibitors, and absorption enhancers to open the tight junction of intestinal epithelial cells and enhance the paracellular transport of proteins only slightly improves the oral bioavailability of insulin. Recently, nanomedicine has been considered a promising tool for the oral absorption of biological drugs, such as insulin, since including drugs in nanoformulations can protect against denaturation and enzymatic or hydrolytic degradation in the gastrointestinal tract [24,38].

Furthermore, nanoformulations can ensure significant drug absorption via the oral route due to their ability to cross the intestinal epithelial barrier via paracellular diffusion [38,39] between the enterocytes and via endocytosis in specialized membranous epithelial (M) cells [40] of the intestinal mucosa (Figure 4).

Various properties of nanoparticles can be tailored to improve trans-epithelial transport mechanisms. Surface modifications through conjugation with ligands for specialized receptors expressed on epithelial cells can enhance receptor-mediated endocytosis. Positively charged groups may facilitate nanoparticle endocytosis via ionic interactions with the negatively charged endothelial cell membranes. Size reduction and hydrophilic coatings may facilitate paracellular diffusion.

Recently, insulin-loaded PLGA nanoparticles functionalized with Fc fragments on their surface were reported to target the neonatal Fc receptor in the intestinal epithelium. Tested in vivo, these Fc-fragment-decorated nanoparticles improved transport across the intestinal epithelium, resulting in a tenfold higher absorption efficiency compared to non-targeted nanoparticles [41]. In a recent study, chitosan-based nanoparticles demonstrated efficacy in T1D animal models [42]. This study showed that oral nanoparticles increased the bioavailability and efficacy of insulin in mice compared to oral free-form insulin. Moreover, the bioavailability of the insulin from the nanoparticles was comparable to that achieved with a subcutaneous administration of free insulin. Indeed, 2 h after treatment, the nanoparticles and subcutaneous insulin provided an equivalent 50%, 20%, and 10% distribution in the liver, small intestine, and kidneys, respectively. Oral free-form insulin, on the contrary, remained in the small intestine. In preparing insulin nanoparticles, commonly used materials include natural polymers such as chitosan, sodium alginate, proteins, hyaluronic acid, etc., as well as synthetic polymers such as PLA, PLGA, etc. Among the other materials available, polysaccharides have demonstrated some advantages, such as excellent biocompatibility, high insulin-loading ability, and easy structural modifications, allowing modulation of the matrix characteristics. Nanoparticles based on polysaccharides were found to provide a good gastrointestinal absorption of insulin, high biocompatibility, and bio-adhesiveness to the intestinal mucus, which enhanced the insulin’s oral bioavailability [43,44]. Yet, despite the increase in bioavailability achieved with these most recent nanoparticles, the oral bioavailability of insulin remains very low and inadequate to satisfy the therapeutic needs of diabetic patients.

The utility of oral insulin nanoparticles may be limited to replacing long-acting insulin injections, while replacing rapid-acting insulin will require more predictable insulin absorption profiles. Therefore, further research is needed to efficiently increase the absorption of nanoparticles across the intestinal epithelium in order to induce the accumulation of nanoparticles in the liver to an extent adequate to provide insulin storage and release, depending on the organism’s needs. If these goals are achieved, oral administration will effectively solve the problems related to daily injections and benefit diabetic patients [24,37].

## 5. Nanoparticles for Gene Therapy

Gene therapy is a technique for relieving the effects of a defective gene by incorporating the exogenous normal gene [45]. Currently, gene therapy is not limited to adding a gene but also includes gene modulation and editing [46,47]. Gene therapy can be considered a new approach in the treatment of diseases [47]. Its therapeutic effect is provided by oligonucleotides such as DNA, RNA, or small interfering RNA (siRNA) rather than conventional drugs [48] (Figure 5a). Once injected into the body, such therapeutic oligonucleotides are expected to penetrate target cells to silence or enhance the expression of defective genes. The DNA or RNA restores the target protein transduction impaired by defective genes, while the siRNA silences the mRNA of faulty genes (Figure 5b).

The administration of naked oligonucleotides would be highly ineffective in gene therapy, as they are rapidly eliminated by endonucleases present in blood and bodily fluids, are phagocytosed by circulating macrophages, and are unable to enter the target cells due to electrostatic repulsion with the negatively charged biologic membranes. Viral vectors such as adenovirus, adeno-associated virus, and lentivirus, or non-viral vectors such as liposomes or nanocapsules, are used to protect the therapeutic oligonucleotides from the environmental conditions and allow their penetration into the target cells [49].

Viral vectors are viruses containing therapeutic DNA or RNA. They are genetically modified to be no longer pathogenic but can transfer their genetic material to target cells. Despite their high efficiency as gene delivery systems, viral vectors can cause problems such as cytotoxicity, inflammation, and immunogenicity, which also decrease their effectiveness after repeated administrations. Non-viral vectors are nanoparticles that can carry DNA, RNA, and siRNA. They have many advantages over viral vectors as they do not cause cytotoxicity, inflammation, and immunogenicity, are easy to prepare, and can deliver high amounts of therapeutic oligonucleotides.

In diabetes, gene therapy has been widely studied for restoring insulin production in pancreatic cells or transfecting insulin genes into other cells, such as the liver, adipocytes, and muscles [47,50,51], with the aim of inducing production of insulin. In gene therapy for insulin production, vectors can be administered via local injection into the pancreas, muscle, or adipose tissue, or injected into the portal vein to reach the liver. Intravenous injection is also possible. In this case, a different biodistribution of nanoparticles compared to viral vectors can make them more active.

Indeed, circulating nanoparticles have been demonstrated to selectively accumulate in body compartments such as tumors and inflamed tissues due to extravasation through pathological discontinuities in their capillaries and in the liver, spleen, and bone marrow [52]. Accumulation in the liver is particularly advantageous for the transfection of liver cells to induce insulin production. Viral vectors, in contrast, do not provide selective accumulations due to their ability to cross blood vessels and distribute throughout the body. This uncontrolled distribution can cause off-target gene delivery with unwanted side effects [53].

In addition to the interest in gene therapy for insulin production, numerous studies have been conducted on the genes responsible for the evolution of T1DM [54,55,56] and T2DM-related genes as possible treatment targets. Approximately 75 independent genetic loci for T2DM have been identified through genetic studies, and several new therapeutic targets have been determined [55]. Genetic loci could significantly impact drug responses and disease incidence and development [56]. It has been found that gene therapy could inhibit the production of nucleotide-binding oligomerization domain-like receptor protein 3 (NLRP3). The inhibition of NLRP3 mitigates inflammation, protecting against pancreatic-cell apoptosis and preventing the development of T2DM in mice [57].

One strategy to restore glucose homeostasis and slow the progression of diabetes is to silence the glucagon receptor using siRNA. Experimental studies in diabetic mice have demonstrated that the delivery of glucagon receptor siRNA using lipid NP technology can result in a significant increase in plasma glucagon and a decrease in blood glucose levels, restoring glucose homeostasis [58]. Alternatively, the gene encoding glucagon-like peptide 1 (GLP-1), a potent insulinotropic hormone, has been delivered via nanoparticles to boost insulin secretion and islet viability [59]. These nanoparticles were obtained using polyethylene imine, a polymer that can complex oligonucleotides. Gene therapy to target hepatic lipid synthesis has been evaluated for alleviating hyperlipidemia and hepatic steatosis in patients with metabolic syndrome. To this aim, lipid nanoparticles carrying siRNA have demonstrated the ability to modify lipid synthesis in hepatic cells [60].

Also, inflammation of the adipose tissue in patients with obesity and T2DM can be a target for gene therapy. Glucan micro-nanoparticles and lipid nanoparticles containing siRNAs have been used to silence genes in inflammatory phagocytic cells [60]. Gene therapy can also modulate the immune response associated with T1DM. Indeed, T1DM results from the inflammation of the pancreatic islets (insulitis) with leukocyte infiltrates, mainly CD8+ T-cells, B-cells, macrophages, and NK cells [51]. Studies have shown that immunosuppressive cytokines such as IFN-γ, TGF-β1, IL-10, and IL-4 can suppress the progression of autoimmune destruction in pancreatic β-cells. Nanoparticles of poly(α-[4-aminobutyl]-l-glycolic acid) containing a plasmid encoding IL-10 were intravenously administrated in NOD mice [61], and the prevalence of severe insulitis was significantly decreased in the nanoparticle-treated mice (15.7%) compared to those injected with naked DNA (34.5%) and the untreated controls (90.9%). This study indicated that this nanoformulation could potentially prevent autoimmune diabetes mellitus.

In another study, plasmid DNA encoding IL-4 and IL-10, loaded in poly[gamma-(4-aminobutyl)-L-glycolic acid] nanoparticles, was intravenously injected into non-obese diabetic (NOD) mice. Six weeks after injection, 75% of the observed islets were intact, compared with less than 3% in the control group. Moreover, only 5% of the islets were severely infiltrated by lymphocytes, compared with over 30% in the control group [62].

Another study used nanoparticles based on chitosan containing a plasmid encoding interleukin-4 (IL-4) and interleukin-10 (IL-10), injected intramuscularly in a diabetic mouse model. The results indicated decreased insulitis in the treated mice, and reduced blood glucose levels were detected [60].

## 6. Nanoparticles for Cell Therapy

Cell therapy in diabetes consists of transplanting functioning pancreatic cells into the pancreas of the diabetic patient. In June 2023, the FDA approved Lantidra (Donislecel). This was the first approval of an allogeneic (donor) pancreatic islet cell therapy using deceased donors’ pancreatic cells for treating type 1 diabetes. Lantrida can be infused into the hepatic portal vein via percutaneous or transvenous transhepatic access, or, if these procedures are not feasible, via laparoscopic access or open surgery. Despite the positive results obtained in clinical trials, Lantidra presents some drawbacks, such as the complexity of its manufacturing process, which requires the purification of pancreatic islets from eligible donors and the need for immunosuppressive drugs that must be administered permanently to prevent islet graft rejection. Using pluripotent or multipotent stem cells to produce surrogate β-cells for transplantation is less cumbersome. To this aim, various types of stem cells can be exploited, including induced pluripotent stem cells (iPSCs), embryonic stem cells (ESCs), and adult stem cells [51,52,53,54,55,56,57,58,59,60,61,62,63]. Technological advancement has facilitated the development of stem cells using different kinds of tissue sources, such as adipose tissue, skin, bone marrow, umbilical cord blood, periosteum, and dental pulp. In searching for promising stem cells, the pancreas is the first organ of choice. Studies with animal models have indicated that a small number of pancreatic tissues, when made available, could restore the optimum pancreatic β-cell mass [51,53,54,55,56,57,58,59,60,61,62,63,64].

To produce surrogate cells, it is necessary to modify the expression of some target genes in selected cells. To this end, stem cells can be transfected with plasmids or edited using CRISPR in vitro, and the modified cells can be subsequently transplanted into patients [51,54,55,56,57,58,59,60,61,62,63,64,65]. Various methods include viral and non-viral vectors, such as calcium phosphate coprecipitation, lipofection, direct microinjection, electroporation, and the biolistic (gene gun) method. Increasing attention is now given to using nanoparticles such as liposomes for in vitro cell transfection to obtain modified cells for cell therapy. Many studies report the successful use of nanoparticles for this purpose. CRISPR has edited pluripotent stem cells isolated from patients for genes with possible mutations. Engineered cells were subsequently differentiated into pancreatic progenitor cells and transplanted into patients [51,54,55,56,57,58,59,60,61,62,63,64,65]. A lecithin-based nano-liposomal carrier was used to encapsulate CRISPR/Cas9 complexes to target the genes associated with diabetes [51,55,56,57,58,59,60,61,62,63,64,65,66]. Cationic lipid-assisted PEG-PLGA nanoparticles were used to encapsulate CRISPR/Cas9 plasmids. After transfection, Cas9 expressed in macrophages disrupted the netrin-1 gene—a therapeutic target for diabetes [26]. In general, encouraging results have been obtained from transplants in treating diabetes. Still, introducing new insulin-producing cells into the body has often generated a foreign-body response, graft rejection, or an innate immune response against these cells [51,57,58,59,60,61,62,63,64,65,66,67,68]. Therefore, long-term protection against these responses is required to ensure the survival and function of transplanted insulin-producing cells.

Nanotechnology has been used to isolate and protect transplanted cells from the immune system while allowing the sufficient diffusion of oxygen, glucose, insulin, and other necessary nutrients [51,58,59,60,61,62,63,64,65,66,67,68,69]. Various coating approaches, including layer-by-layer polymer deposition and chemical reactions of polymers, have been applied to islets to produce nano-thin coatings that can protect the activity of islets without inhibiting their function [51,60,61,62,63,64,65,66,67,68,69,70,71]. However, the lack of encapsulating materials that can avoid host recognition and subsequent foreign-body responses severely limits the translation of these technologies into the clinic [51,60,61,62,63,64,65,66,67,68,69,70,71]. Future advancements in islet encapsulation require the development of materials and devices that allow the encapsulated cells to maintain their function and viability while avoiding fibrosis [72,73].

## 7. Nanoparticles in Diabetic Wound Healing

One of the most troublesome complications for diabetic patients is diabetic wounds, which have emerged as one of the most prominent threats to human health worldwide [74]. Diabetic wounds can develop into chronic intractable ulcers due to persistent infection caused by cellular dysfunction, microcirculatory disorders, high levels of oxidative stress, and hypoxia, which are the leading causes of amputation and disability [75].

The traditional clinical treatments include controlling blood glucose levels, surgical debridement, skin transplantation, wound dressing, and hyperbaric oxygen therapy. Although these clinical treatments can achieve symptom control, they have a limited therapeutic effect on diabetic wound healing. Thus, there is an urgent need for treatments that are potentially effective, painless, and scar-free for diabetic wounds. Nanoparticles have long been used as excellent biologically active delivery systems for wound healing since they can carry many drugs with different physicochemical properties, easily absorb cells, have good biocompatibility, and effectively control drug delivery over time [74,75,76,77].

These properties make nanoparticles suitable for wound healing. They can effectively release active substances in the pathological environment, enhancing angiogenesis, collagen production, and extracellular-matrix deposition, thus favoring tissue repair. Epidermal growth factor (EGF) and vascular endothelial growth factor (VEGF) are the most active substances in wound repair. In diabetes wounds, EGF and its receptor are lacking, resulting in less matrix deposition, poor tissue-repair function, and complex wound healing. Naked EGF has been widely administered in the treatment of diabetic wounds, but its utility has been hampered by its short half-life. Nanoparticles of PLGA containing EGF have demonstrated improved efficacy concerning free EGF and promoted fibroblast proliferation. Nanoparticles based on carboxymethyl chitosan have demonstrated improved efficacy due to the protection of EGF from environmental hydrolysis [78,79,80,81].

The administration of VEGF via local injection or topical application has been demonstrated to enhance wound healing. However, the low stability of VEGF in vivo and its degradation by proteases in the wound bed strongly limit its use as an unrestricted substance, requiring high dosages and frequent administration to provide a therapeutic effect [82].

Nanoparticles of PLGA containing VEGF were studied on non-diabetic and diabetic wounds, and it was demonstrated that they enhanced the proliferation and migration of keratin-forming cells and upregulated VEGFR2 expression at the mRNA level [82]. Other active molecules incorporated in nanoparticle formulations have been evaluated in wound healing. Curcumin is a natural hydrophobic polyphenol with anti-inflammatory, antioxidant, antibacterial, and antitumor effects. However, its therapeutic effect is hindered by its low bioavailability and poor stability in bodily fluids [73,83]. Chitosan nanoparticles loaded with curcumin were evaluated in a diabetic model, demonstrating improved skin wound-healing rates, granulation tissue formation, collagen deposition, and VEGF production [84]. 

Sesamol is a phenolic compound with antioxidant, anti-inflammatory, and anti-hyperglycemic properties. PLGA nanoparticles containing sesamol accelerate wound healing by increasing ECM deposition and downregulating the expression of inflammatory mediators [85]. Melatonin is an indoleamine with a reactive oxygen species (ROS)-scavenging effect. However, it is easily oxidized and poorly water-soluble. Melatonin-loaded chitosan nanoparticles were evaluated for healing via topical application in diabetic rats. This treatment improved wound healing by increasing fibroblast and angiogenic proliferation and enhancing collagen deposition [86].

Ferulic acid has been reported to have dual properties, i.e., anti-diabetic and antioxidant properties. Ferulic acid-poly (lactic-co-glycolic acid) nanoparticles, topically administered in the wounds of diabetic rats, induced a faster epithelization compared with the diabetic wound control group in one study [87].

All-trans retinoic acid (ATRA) stimulates fibroblast proliferation and reduces MMP expression. These characteristics have generated interest in its use to promote skin wound healing. However, its poor water-solubility prevents its use as a free drug. Nanoparticles have been prepared with chitosan and a lipid core containing ATRA to improve ATRA’s bioavailability. These systems have demonstrated increased collagen deposition in diabetic wounds [88,89,90].

## 8. Conclusions

The use of nanomedicine holds promise in the management of diabetes and its complications. Many preclinical studies have demonstrated the potential of nanoformulations to improve the response rate to diabetes treatments. However, clinical studies are still needed to fully evaluate their outcomes over long-term treatments and allow their transition to clinical use. Until now, the Food and Drug Administration has approved only one liposome-like nanoformulation for insulin, HDV-I, but it has not yet been marketed due to various adverse effects that emerged in clinical trials.

Among the different nanoformulations, glucose-responsive nanoparticles are regarded with high interest as they activate insulin release in response to high glucose levels. These nanoparticles have been proven to control blood glucose over time, thus avoiding frequent insulin injections, reducing the lag between sensing and insulin delivery, and providing an overall increased response rate to changes in glucose levels.

Nanoparticles for the oral administration of insulin represent a long-sought alternative to the frequent SC injections that diabetic patients must undergo, which cause discomfort, pain, suffering, and the possibility of local infections. Moreover, oral insulin pharmacokinetics are like the pharmacokinetics of endogenous insulin, thus allowing the modulation of insulin blood levels similar to physiological secretion.

Nanoformulations for gene delivery and cell therapy may stimulate insulin production in β-cells or β-like cells through controlled gene expression or cellular reprogramming. The modification of gene expressions responsible for the evolution of T1DM or T2DM or the modulation of immune cell responses can also be achieved. Nanomedicine for the treatment of diabetes could have a great effect on patients with this disease, but in the future, it will be necessary to discuss the ethical and social implications of advanced nanomedicine and gene therapy, including issues such as accessibility, affordability, and potential inequalities in healthcare.

## Figures and Tables

**Figure 1 ijms-25-07028-f001:**
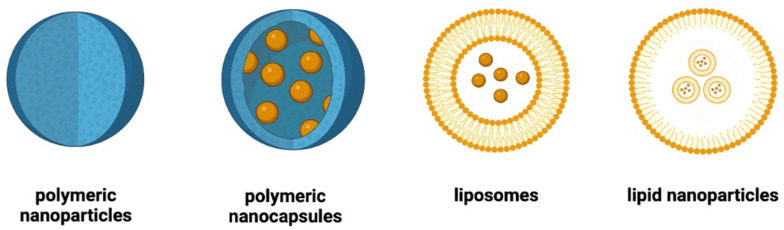
Schematic representation of the main types of nanoparticles used in the management of diabetes.

**Figure 2 ijms-25-07028-f002:**
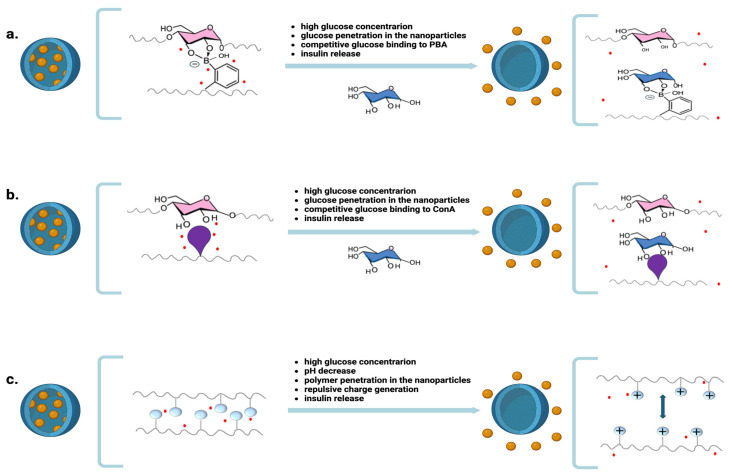
Mechanisms of insulin release from nanoparticles in response to increased glucose levels in blood and bodily fluids: (**a**) Environmental glucose penetrates the polymeric matrix. It links PBA in place of the glucose monomer of the matrix. This generates a loss of the polymeric structure and a consequent insulin release. (**b**) Environmental glucose penetrates the polymeric matrix. It links ConA in place of the glucose monomer of the matrix. This generates a loss of the polymeric structure and a consequent insulin release. (**c**) The increased environmental glucose decreases the pH, thus inducing a protonation of the polymeric chains in the nanoparticle matrix. This generates a repulsive forces that disassembles the matrix and allows insulin to be released.

**Figure 3 ijms-25-07028-f003:**
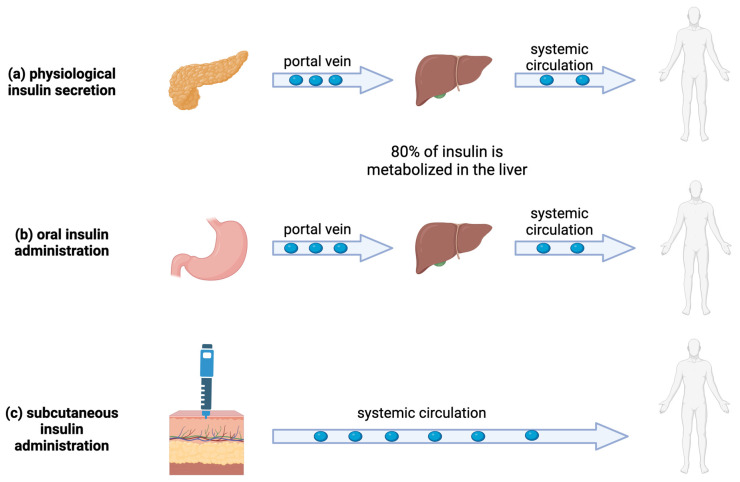
Insulin secretion and body distribution: (**a**) In physiological conditions, insulin is secreted by pancreatic islets into the portal vein, reaches the liver, and subsequently enters the systemic circulation. (**b**) With oral administration, insulin is absorbed by the intestinal mucosa capillaries, which transport it to the portal vein and to the liver, from which it reaches the systemic circulation. (**c**) With subcutaneous administration, insulin is absorbed by the epidermal capillaries and reaches the systemic circulation.

**Figure 4 ijms-25-07028-f004:**
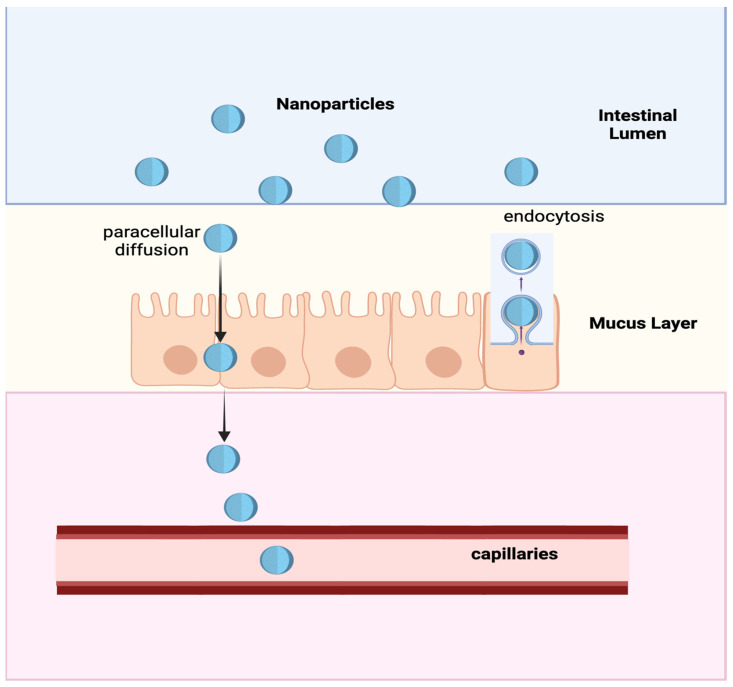
Schematic representation of the main mechanisms of nanoparticle absorption through the intestinal mucosa: Paracellular diffusion occurs between endothelial cells and endocytosis in M-cells (or microfold cells), whose name derives from their particular composition. M-cells are specialized intestinal cells and are part of GALT lymphoid follicles such as Peyer’s patches. Their function is to transport antigens from the luminal side to the sub-epithelium (transcytosis). This is possible due to an exclusive cellular structure, including many basolateral membrane invaginations that enable macrophages and other immune cells to begin an immune response. Transcytosis is also being explored as an intestinal drug and vaccine delivery opportunity.

**Figure 5 ijms-25-07028-f005:**
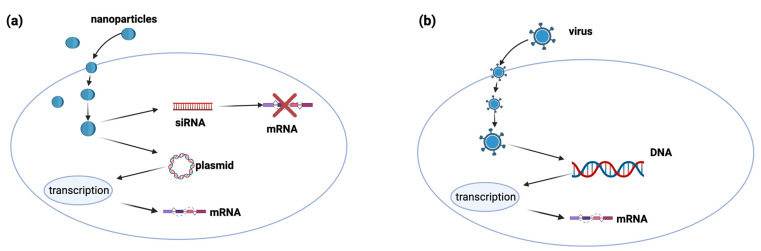
The basic mechanisms of gene delivery. (**a**) Non-viral gene delivery: Nanoparticles carrying siRNA or plasmids enter the cells via endocytosis, releasing their content in the cytoplasm. siRNA links homologous mRNA, inducing its degradation and the consequent repression of target proteins. Plasmids translocate to the nucleus and induce specific mRNA transcription, thus increasing target protein production. (**b**) Viral gene delivery: Viruses carrying specific DNA sequences enter the cells via endocytosis and then release their content in the cytoplasm. DNA translocates to the nucleus, inducing specific mRNA transcription, thus increasing target protein production.

## Data Availability

Not applicable.

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
