# Peer review of "Nanomedicine in the Treatment of Diabetes"

_ijms, 2024, doi:10.3390/ijms25137028_

Round 1

Reviewer 1 Report

Comments and Suggestions for Authors

The Paper entitled "Nanomedicine in the Treatment of Diabetes" describes very interesting field in diabetes research, the Authors provide information about crucial areas in which nanomedicine can be used in diabetes treatmend and make it in a well-organized way. My general opinion on this manuscript is very positive. 

Nevertheless, there are some flaws that need to be corrected: 

Major flaws: 

1. The Authors don't desrcibe the shortcomings of nanomedicine applications and potential limiting factors. For instance - in 2.2. Chapter the Authors write about the problem with PBA specificity to glucose or other saccharides, but there is lack of information wheter PBA is specific only to saccharides (or that PBA bidnings to other compounds in human plasma can be neglected). 

2. The Authors only gave a general view upon pre-clinical and clinical studies. For example - if there are any clinical studies upon nanomedicine application described, the information should be provided (at least the most important examples, not every clinical study in every nanomedicine application described in the manuscript). At the end of chapter 2.2. there are sentences - "There are many preclinical studies" and, finally "there is no FDA approvement". It should be written if there are clinical studies or this application has only been investigated on preclinical stages. 

3. The Authors didn't give quantitative data, for example - in Chapter 2.3. it would be precious if the information about absorbtion percentage with nanocapsules vs without nanocapsules (e.g. in cellular Caco2 cells, if such studies exist) is provided. Moreover, in Chapter 2.4. (gene therapy) some quantitative data would be warranted (e.g. line 353 - "(...)The study showed an evident reduction" - the data number or percentage should be provided.)

Minor faults: 

1. It is a Review, therefore the section "Discussion" should not be written, 2.1 should be Chapter 2, 2.2 - Chapter 3 and so on. 

2. In the section 2.1. the nanoparticles classification on 4 types is provided, but in the text the information on these types overlaps, it should be separated in a more unequivocal way.

3. Figure 3 - the information above the liver image "80% of insulin is metabolized" should be provided. 

4. Figure 5 - at the bottom of both circles the "mRNA", not "miRNA" should be written. 

5. First sentence lin line 308 (i.e. "Production in other cells, such as liver, adipocytes, and muscle [50 -52"]) is unclear.  

Author Response

We thank the reviewer for his constructive comments, and we follow the advice given. Specifically:

Major flaws: 

  • We thank you for your comment and we reported in the text: “one problem with PBA is the need for more specificity for glucose, since as it has the ability to interact with diols from other molecules than glucose present in the body such as other carbohydrates, catechol, nucleic acids, low molecular weight proteins. Efforts are underway to develop PBA derivatives that may have greater specificity for glucose at physiological pH [27, 28].”
  • We thank you for your comment and report in conclusion: “Until now, the Food and Drug Administration has approved only one liposome—like nano-formulation for insulin, HDV-I, but it has not yet been marketed due to various adverse effects that emerged in clinical trials.”
  • At the end of chapter 2.2. (now Chapter 3) we reported, “Currently, only preclinical studies have been carried out on insulin therapeutics whose bioactivity is regulated by blood glucose levels. Further research is needed to make these systems suitable for entering clinical trials.”
  • We thank the reviewer for his comment. In chapter 2.3 (now Chapter 4), Quantitative data regarding bioavailability and biodistribution, which are directly related to absorption, have been reported for nanoparticles compared to subcutaneous and oral insulin.
  • In Chapter 2.3 (now Chapter 5) Quantitative data have been reported instead of the evident reduction

Minor flaws: 

  • We thank the reviewer for his suggestion, and we have change the number of the chapters/paragraphs as adivced
  • We have separated in the text the information on the different types of Nanoparticles
  • We agree with the reviewer, please find the figure 3 corrected as adviced
  • We agree with the reviewer, we have correct the figure
  • We thank the reviewer for his comments, we have correct the sentence

Reviewer 2 Report

Comments and Suggestions for Authors

General comments:

This is an interesting paper that covers several topics related to potential future uses of nanotechnology in diabetes care and management. While each section could be a review article in itself, this review would serve as a good starting point for someone new to the field.

There are many paragraphs that only have one sentence, the authors need to work on tying these ideas together into coherent paragraphs, which should consist of at least 3 sentences in the reviewers opinion.

Introduction:
In the second paragraph of the introduction, no mention of oral anti-diabetic agents is made.  A significant number of patients with T2D use oral medications, so leaving this out misrepresents reality.

Page 2 Line 51: The sentence “They are now actively exploited….” Is confusing to the reviewer, can the authors provide examples of where nanotechnology is being used in clinically approved treatments?  There is a difference between a treatment having been published in a journal, and it being used in treatment of diabetes not within a clinical trial.  (This idea needs to be corrected in other places as well, Figure 2 caption, etc.). It does seem that in other locations the authors realize this difference, such as in the conclusion, so this needs to be made consistent thru the paper.

Author Response

We thank the reviewer for his comments. We have provided corrections to the paper as suggested. We agree with the reviewer regarding the sentences in each paragraph, and we have tied more sentences in the paragraphs together with a few sentences.

  • Introduction: we have reported the main oral anti-diabetic agents in the Introduction.
  • We corrected the confusing phrases throughout the paper as suggested. In particular: we substituted “actively exploited” with “actively studied.” At the end of chapter 2.2. (now Chapter 3) we reported that no clinical studies are performed now for the systems described, and only preclinical studies have been performed. In conclusion we reported that until now the Food and Drug Administration has approved only one liposome- like nano delivery system for insulin, HDV-I. However, it has not been marketed due to the various adverse effects of drugs in different clinical trials.

Reviewer 3 Report

Comments and Suggestions for Authors

The review “Nanomedicine in the Treatment of Diabetes” by Andreadi et al. provides a comprehensive overview of the applications of nanomedicine in diabetes treatment, including both Type 1 and Type 2 diabetes. Maintaining normoglycemia is critical for both T1DM and T2DM patients to avoid severe complications such as blindness, kidney and heart disease, nerve damage, and increased infection risk. Current treatments include frequent blood glucose monitoring and insulin administration. Nanomedicine offers promising advancements for diabetes management, utilizing nanoparticles for controlled drug release, protection of insulin from degradation, and targeted gene and cell therapy.

The review highlights the potential of different nanotechnological approaches to improve therapeutic outcomes, such as glucose-sensitive nanoparticles for insulin delivery and nanoparticles for gene therapy and wound healing. I found this review very interesting and well structured. The analysis of the different approaches has been done in a clear and comprehensive manner.

However, the Authors should further discuss the ethical and social implications of advanced nanomedicine and gene therapy, such as accessibility, affordability, and potential inequalities in healthcare.  Addressing these aspects would provide a more comprehensive perspective.

In addition, the text in all the figures needs to be enlarged as it is difficult to read.

Author Response

We thank the reviewer for his comments regarding the paper and we have modify the article and the figures as suggested. We also agree regarding the social implications and we have add a comment in the conclusions.